# Brain responses to food viewing in women during pregnancy and post partum and their relationship with metabolic health: study protocol for the FOODY Brain Study, a prospective observational study

Anna Lesniara-Stachon [1,2] Dan Yedu Quansah,[1] Sybille Schenk,[1,3]
Chrysa Retsa,[4,5,6] Ryan J Halter,[7,8] Micah M Murray,[4,5,6] Alain Lacroix [9,10]
Antje Horsch [9,10] Ulrike Toepel,[4] Jardena J Puder [1]

**Correspondence to**
Professor Jardena J Puder;
Jardena.Puder@chuv.ch

## ABSTRACT

**Introduction** Food cravings are common in pregnancy and along with emotional eating and eating in the absence of hunger, they are associated with excessive weight gain and adverse effects on metabolic health including gestational diabetes mellitus (GDM). Women with GDM also show poorer mental health, which further can contribute to dysregulated eating behaviour. Food cravings can lead to greater activity in brain centres known to be involved in food 'wanting' and reward valuation as well as emotional eating. They are also related to gestational weight gain. Thus, there is a great need to link implicit brain responses to food with explicit measures of food intake behaviour, especially in the perinatal period. The aim of this study is to investigate the spatiotemporal brain dynamics to visual presentations of food in women during pregnancy and in the post partum, and link these brain responses to the eating behaviour and metabolic health outcomes in women with and without GDM.

**Methods and analysis** This prospective observational study will include 20 women with and 20 without GDM, that have valid data for the primary outcomes. Data will be assessed at 24–36 weeks gestational age and at 6 months post partum. The primary outcomes are brain responses to food pictures of varying carbohydrate and fat content during pregnancy and in the post partum using electroencephalography. Secondary outcomes including depressive symptoms, current mood and eating behaviours will be assessed with questionnaires, objective eating behaviours will be measured using Auracle and stress will be measured with heart rate and heart rate variability (Actiheart). Other secondary outcome measures include body composition and glycaemic control parameters.

**Ethics and dissemination** The Human Research Ethics Committee of the Canton de Vaud approved the study protocol (2021-01976). Study results will be presented at public and scientific conferences and in peer-reviewed journals.

## STRENGTHS AND LIMITATIONS OF THIS STUDY

⇒ This prospective observational study fills the knowledge gap regarding brain responses to food viewing during and after pregnancy.
⇒ This study includes subjective (questionnaires) and objective methodologies (electroencephalography (EEG), Auracle device, Actiheart) to investigate brain responses to food viewing, eating behaviours and stress responses during and after pregnancy.
⇒ The study will link the outcomes of physiological and behavioural measures.
⇒ We will compare eating behaviours between healthy women and women with gestational diabetes, both cross-sectionally and prospectively.
⇒ Limitations include a small sample size (but comparable to other EEG studies with a similar design) and fact that all analyses (and the correlation of their outcomes) are exploratory, given the lack of data in pregnancy or the post partum.

## INTRODUCTION

### Metabolic health and eating behaviours

The prevalence of obesity continues to increase worldwide.[1] In the USA, 35.7% of women of reproductive age (20–39 years) are obese[2] and this prevalence exceeds 20% in high-income western countries in general.[3]

The widespread and unrestricted access and intake of foods high in sugar and fat contribute to the obesity epidemic. Foods high in sugar (carbohydrates) and fats, such as chocolate, sweet desserts and salty snacks are the most frequently craved foods.[4] Sugar strongly stimulates the reward system, which can trigger compulsive eating.[5] Eating behaviour refers to a variety of eating practices. This includes cognitive restraint,

disinhibition with high responsiveness to external food cues and susceptibility to hunger.[6] Disinhibited eating behaviour reflects a tendency to overeat. This can happen in the context of negative emotions (such as anxiety, depression, anger and loneliness) and/or in response to a high food palatability.[7] Disinhibited eating behaviours can be associated with compulsive intakes of high energy-dense foods rich in fat and sugar, which contribute to obesity and poorer long-term metabolic health.[7] The ideal situation whereby eating behaviours are regulated by appetite and food consumption is challenged by disinhibited eating behaviours.[8]

Another aspect of eating behaviour that relates to metabolic health is eating rate. Studies showed that faster eating rate is associated with increased food intake[9] and an increased risk of obesity.[10] A meta-analysis of the cross-sectional and longitudinal association between eating speed and metabolic outcomes showed that faster eating rate is associated with higher risks for the metabolic syndrome, central obesity, elevated blood pressure and increased fasting plasma glucose compared with slower eating.[11]

### Brain regions in relationship to food cues

Although the underlying cause(s) of food cravings are not deeply understood, it has been shown in the general, non-pregnant population that cue-induced cravings in response to food pictures predict disordered eating and excess weight gain. In a meta-analytical review of 45 studies, exposure to food stimuli (pictures) and the experience of craving influence eating behaviour and further weight gain independently of baseline body mass index (BMI), age and gender.[12]

Some people are more susceptible to an obesogenic environment. This can be also mediated through different brain responses to food in key areas. The few existing clinical data show that reward-related brain regions may be involved in excess weight gain.[13] 'The reward circuit is a complex neural network' (the cortico-ventral basal ganglia circuit), which consists, among other, of the orbitofrontal cortex, anterior cingulate cortex and the ventral striatum which is part of the basal ganglia.[14] The ventral striatum receives information from the orbital prefrontal cortex, the insular cortex, the cingulate cortex and also the amygdala, which is also closely linked to sensory processing.[14] These reward-related brain regions are stimulated when viewing the craved foods and through unconscious processes of learning/conditioning mechanisms, promote further greater 'wanting' and seeking of these foods.[15] This could be related to the fact that higher intake of foods rich in fat and sugar can reduce reward thresholds in the limbic striatal pathways and the amygdala.[15–17] This then promotes further food craving, further increases the neural reactivity to food cues in reward-related brain regions even more, and can thus promoting a higher intake of palatable foods whether in a hungry or satiated state.[15 16 18] Investigating the brain's response to food with different fat and carbohydrate content may

help to understand these mechanisms and their relationship to eating behaviour and metabolic health. So far, this has been mostly studied in healthy subjects.

Furthermore, physiological and metabolic parameters are also related to how the brain reacts to the sight of food. Response to viewing high-energy and low-energy foods differ between obese and healthy weight subjects.[19] In an functional MRI (fMRI) study, overweight and obese participants demonstrated greater activation in reward-related brain regions, like the ventral striatum and anterior cingulate compared with normal weight subjects.[19] Insulin resistance and type 2 diabetes mellitus (DM2) are associated with changes in brain responses to food stimuli.[20] In patients with DM2, cross-sectional studies using fMRI revealed that compared with controls, responses to viewing food pictures are stronger in the ventral striatum and cortical regions, such as the insula and the orbitofrontal cortex, which are related to reward-related circuits, in comparison to controls.[20]

### Stress, emotions and eating behaviour

Emotions and stress can influence food choices and eating behaviour.[21 22] Studies showed that there is a link between negative mood, stress and the intake of high-fat, high-carbohydrate-hyperpalatabable food and higher overall food intake, which contribute to an increased risk for obesity.[22 23] Food cravings and overeating in response to stress and negative mood are particularly relevant in people who engage in more frequent emotional eating.[23–25] In an fMRI study, people who engage in more frequent emotional eating had stronger responses to high-calorie food in reward-associated areas (anterior cingulate cortex) when being in a negative mood in comparison to people who do not engage in more frequent emotional eating.[26]

These results together demonstrate that there is a link between brain responses to food viewing and metabolic health, but also stress and emotions and that reward-related brain regions play a role in the regulation of food intake.

### Pregnancy

Maintaining a healthy body weight is particularly important during pregnancy, as it impacts on the health of the mother and her offspring.[27] Maternal obesity and overnutrition during pregnancy affect the infant's metabolic programming and increase its risk of obesity, type 2 diabetes and cardiovascular disease in the future.[28]

Nevertheless, around 28% of women gain more weight than recommended[29] and excessive gestational weight gain (GWG) is associated with adverse perinatal outcomes such as pre-eclampsia, hypertension in pregnancy, gestational diabetes, preterm birth, caesarean section and small or large size for gestational age at birth.[30] Higher GWG is also associated with greater postpartum weight retention (PPWR) and increasing the risk for future diabetes.[31]

During pregnancy, there is a change in eating behaviour that can have long-term effects for the mother's metabolic

health.[32] Food cravings are frequent and concern around 40% of pregnant women.[33] They typically begin at the end of the first trimester, peak in their frequency and intensity in the second and third trimester, and decrease after delivery.[34] Higher food cravings during pregnancy have been associated with higher mean energy intake and GWG.[33]

Emotions and stress also influence eating behaviour in pregnancy.[35] Emotional cues such as negative affect can trigger the craving for high-fat foods in pregnant women which is associated with excessive GWG.[8] Similarly, under stress, pregnant women are more likely to increase cravings for pleasurable food.[36] There are no data regarding brain responses to food during pregnancy (using either MRI or electroencephalography (EEG)) or regarding the impact of emotions or stress on these brain responses.

Regarding metabolic health, insulin resistance increases in pregnancy and around 16% of pregnant women worldwide develop gestational diabetes mellitus (GDM).[37] GDM is defined as glucose intolerance with the first onset during pregnancy without fulfilling the criteria of overt diabetes.[38] Women with GDM are at higher metabolic risk, have more perinatal complications such as pre-eclampsia or caesarean section[39] and also have a higher prevalence of mental health problems in pregnancy such as depression.[40]

### Postpartum period

There is a lack of data regarding eating behaviours in the post partum and its relationship to PPWR.[32] Existing studies showed that higher restrained and also higher intuitive eating are associated with postpartum weight loss.[32] Knowledge related to cravings and emotional eating in the post partum is limited. Negative emotions play a role in eating behaviour during this period, leading to increased food intake and contributing to PPWR.[41] Regarding the group of women of higher metabolic risk, those with GDM have more adverse maternal health outcomes in the post partum such as higher PPWR, a 7–10 fold higher risk of diabetes[42] and an increased risk for cardiovascular disease.[43 44] GDM is also associated with higher risk of post partum depression[45] which is also associated with a less healthy diet.[46]

### Electroencephalography

EEG and visual evoked potentials (VEPs; patterns of electrical activity in the brain in response to a visual stimulus) to food viewing offer the advantage of disentangling the temporal as well as spatial dynamics of perception, while fMRI only unravels spatial aspects, that is, brain regions. Some VEP studies have shown differential responses to pictures of high and low-energy foods.[47] Reward-related and (inhibitory) control-related processes are partially recruiting the same neural sources, yet with varying temporal dynamics.[48] Existing studies on food cravings during and after pregnancy only assessed different kinds of food cravings and their frequency with questionnaires and did not involve brain responses to food cues.

Results from these studies have been inconsistent. We are not aware of any human studies investigating objective correlates or mechanistic pathways of cravings in pregnancy. In order to better understand food cravings and metabolic health outcomes in pregnancy and post partum, investigating the neural activities of the brain pathways in response to cravings induced by food viewing during and after pregnancy might lead to novel clinical treatment options.

Overall, the relationship between weight, GWG, PPWR and adverse perinatal and metabolic health outcomes in the perinatal period and the difficulty to reduce weight or weight gain, highlight the need for novel approaches to understand eating behaviour and metabolic health during pregnancy and the post partum, especially in metabolic high-risk women with GDM.

As there are no data regarding brain responses to food in pregnancy and in the post partum, we designed an observational prospective study whose design is exploratory. Nevertheless, based on the above-mentioned data, we hypothesised that brain responses, especially to food high in fat and carbohydrate content would differ between pregnancy and the post partum and between healthy women and women at higher metabolic risk such as GDM. We also hypothesised that there would be a relationship between brain responses to viewing food with different fat and carbohydrate content, eating behaviour and metabolic health and that negative emotions and higher stress levels might influence these brain responses as well as eating behaviour in the perinatal period.

### Aims of the present study

The overall aim of this observational study is to investigate brain responses to viewing pictures of foods of different fat and carbohydrate contents in women during pregnancy and in the post partum and their relationship with eating behaviour and metabolic health. We will investigate this in healthy women and in a metabolic high-risk group, that is, women with GDM to include a wide spectrum of metabolic health.

The main objectives are as follows:

Objective 1: To investigate differences in brain responses to food pictures of varying fat and carbohydrate content (HIGH-FAT/LOW-CARB; HIGH-FAT/HIGH-CARB; LOW-FAT/LOW-CARB; LOW-FAT/HIGH-CARB) during and after pregnancy, in women with and without GDM.

Objective 2: To investigate the cross-sectional and longitudinal relationships between brain responses to food viewing with reported food cravings, eating behaviour and metabolic health outcomes such as weight, body fat and glucose control during pregnancy and in the post partum.

Objective 3: To investigate differences in brain responses to food pictures in relationship to different emotional states and stress levels.

## METHODS

### Study design

This is a prospective observational study of women with and without GDM. There will be two visits (at 24–36 weeks gestational age and 6 months post partum). Both visits will take place at Lausanne University Hospital.

### Study population, inclusion and exclusion criteria, and recruitment

We will include 40 women with valid data for the primary outcomes (20 women with GDM and 20 healthy controls) at both time points. In the initial tests, the EEG data of 14 women with GDM and 13 healthy controls were not valid (ie, too much noise, low quality of signal), which has been corrected since. We will therefore recruit at least 35 women with GDM and at least 34 healthy controls, in their 24–36 weeks of pregnancy, during their antenatal appointments visits at a Swiss University Hospital. GDM women (clinical group) will be recruited at the GDM clinic whereas healthy pregnant women without GDM (control group) will be recruited at the hospital's ambulatory clinic.

For women with GDM, inclusion criteria are: gestational age 24–36 weeks, age ≥18 years, GDM diagnosed at 24–32 gestational weeks according to IADPSG and ADA criteria.[49 50] For healthy controls, inclusion criteria include women with normal weight (prepregnancy BMI <25 kg/m2), with no prior or current GDM, no diets and weight changes above 5 kg in the last 5 years (except for pregnancy), no excessive pregnancy weight changes (less than twice of the Institute of Medicine recommendations).[51] Women in both groups should be able to fluently speak and write either French or English.

Exclusion criteria include: pre-existing diabetes, eating disorders (Anorexia Nervosa, Bulimia Nervosa and Binge-Eating Disorder), vegetarian/vegan diet, uncontrollable nausea and vomiting, current insulin treatment, participation in an intervention study, current medication to treat psychiatric disorders, active suicidal thoughts.

### Primary outcome

The primary outcomes are EEG viewing responses, specifically to food images depending on fat and carbohydrate content (HIGH-FAT/LOW-CARB; HIGH-FAT/

**Table 1** Study schedule

| Flowchart of the study procedure | | | Visit 1 | Visit 2 |
|---|---|---|---|---|
| **Study schedule** | | | | |
| Time point | | | 3rd trimester of pregnancy | 6 months post partum |
| Duration | | | Televisit: 10–30 min for medical history and 20–40 min for online questionnaires Visit: max 2 hours 30 min including breaks* | Televisit: 10–30 min for medical history and 20–40 min for online questionnaires Visit: max 2 hours 30 min including breaks* |
| **Procedures and time** | | | | |
| Televisit before or after each of both of the physical visits (at home) | | | | |
| Medical history if lacking in the chart and questions regarding lifestyle, by phone | | 10–30 min | | |
| Questionnaires online: Food Craving Questionnaire-Trait-reduced, Edinburgh Postnatal Depression Scale, Edinburgh Handedness Inventory, TFEQ-R18, Intuitive Eating Scale-2 | | 20–40 min | X | X |
| During the visit (in the EEG lab up to the break) | | | | |
| Setup EEG, questionnaires: Perceived Stress Scale-10 Food Craving Questionnaire-State PANAS questionnaire: Positive and Negative Affective Schedule, current hunger level: Likert scale) + heart rate (Actiheart) | 13:30–14:15 hours | 30–45 min | X | X |
| Electroencephalography | 14:15–15:10 hours | 55 min | X | X |
| Break | 15:10–15:20 hours | 10 min | X | X |
| Bioimpedance and weight | 15:20–15:30 hours | 10 min | X | X |
| Positive and negative affective schedule current hunger level: Likert scale Break and Auracle* + questionnaire: Eating of the provided snacks and three questions regarding eating duration | 15:30–15:45 hours | 15 min | X | X |
| Height, blood pressure | 15:45–15:50 hours | 5 min | X | X |
| Capillary measure: glycated haemoglobin (HbA1c) Double check lunch composition | 15:50–15:55 hours | 5 min | X | X |
| Set-up glucose sensor: Freestyle Libre 1 (only for GDM) × 14 days | 15:55–16:00 hours | 5 min | X | X |

*During the Auracle session, patients will eat different snacks that we provide.
EEG, electroencephalography; GDM, gestational diabetes mellitus; TFEQ-R18, Three-Factor Eating Questionnaire-Revised 18.

HIGH-CARB; LOW-FAT/LOW-CARB; LOW-FAT/HIGH-CARB). These responses will be examined in terms of amplitude and latency of the brain responses (VEPs) as well as in terms of response topography. An estimation of the underlying intracranial sources of the VEPs will also be performed. The full analysis pipeline is referred to as 'Electrical Neuroimaging' as has been developed and validated by members of our team and their collaborators (eg, Michel and Murray, 2012; Murray et al., 2008; Brunet *et al*, 2011[52]). VEPs will be compared during (visit 1) and after (visit 2) pregnancy, as well as between women with and without GDM.

### Secondary outcomes

Secondary outcomes are weight, glycated haemoglobin (HbA1c), glucose measures at visits 1 and 2 (the latter only in women with GDM) and body composition (fat/lean mass). We will investigate eating rate, objectively using a novel eating recognition device (Auracle) and subjectively (reported perceived eating rate). Using validated self-report questionnaires, we will assess eating behaviours, including cravings (Food Craving Questionnaire-Trait-reduced), intuitive eating (ie,) (The French Intuitive Eating Scale-2 (IES-2)), external, emotional eating and cognitive restriction (The Three-Factor Eating Questionnaire). Depression scores will be assessed by Edinburgh Postnatal Depression Scale and current mood by PANAS questionnaire Positive and Negative Affective Schedule. Other measures will include self-reported stress level assessed with a questionnaire (The Perceived Stress Scale) and objectively measures of stress using Actiheart.

### Data collection and visits

The study started in March 2022 and the end of the study is scheduled for June 2024.

The first visit will take place during pregnancy, at 24–36 weeks gestational age (visit 1) and the second visit at 6 months postpartum (visit 2).

After signing the informed consent, and before visit 1 and visit 2, participants will complete validated questionnaires online (see Questionnaires related to eating behaviour and mental health). They will be contacted by phone to complete missing data regarding their medical history as well as their lifestyle behaviours.

Before each visit, participants will be asked to eat a balanced meal around 12.00 hours (based on the Swiss Society for Nutrition recommendations; the patients are given examples and at the end will be checked what they eat) to minimise the bias of different satiety level, will come to the hospital at 13:30 hours and testing will be completed at 16 hours (table 1).

### Patient and public involvement

The development of the research question and outcome measures was also based on remarks from patients that were followed in clinical settings and we informally asked their advice while designing the study. Some results will be provided to participants at the end of their study as

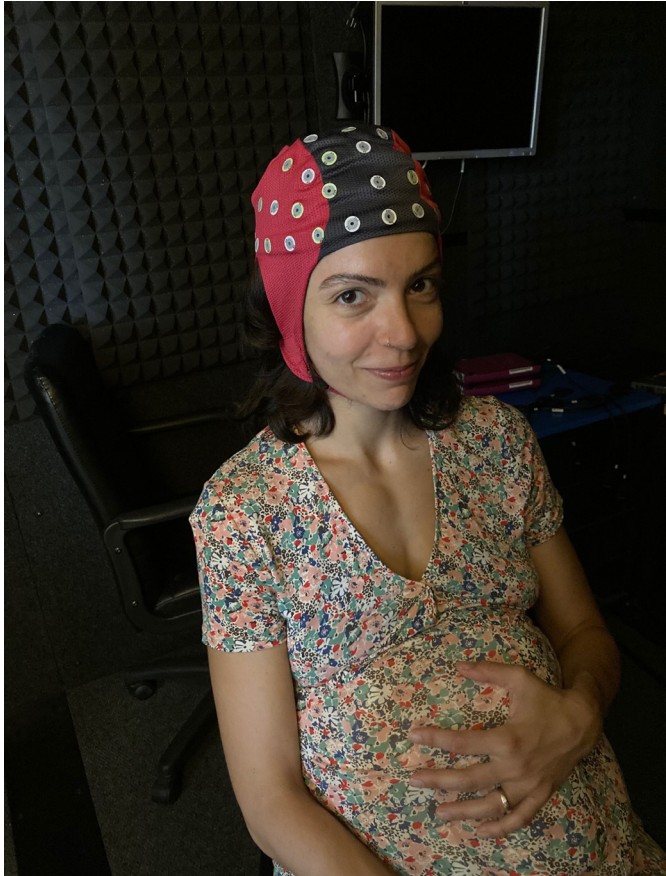

**Figure 1** 128-channel electroencephalography.

feedback by sending them to their healthcare providers. Interested patients will also receive an overall return of the results.

### Measures
#### Behavioural task/EEG

We will employ a behavioural task involving food pictures of variable fat and carbohydrate content and will record high-density VEPs (brain responses to visual stimuli via EEG) (figure 1). Using the Salzburg food picture database,[53] we are going to show 4 blocks of 240 randomised images (200 pictures of food and 40 non-food objects together; 50 pictures of food and 10 non-food objects per block) to the participants. The food images are divided into four categories: HIGH-FAT/LOW-CARB; HIGH-FAT/HIGH-CARB; LOW-FAT/LOW-CARB; LOW-FAT/HIGH-CARB.

HIGH-FAT was defined by a fat content of >5 g/100 g (21.6±13.77 g/100 g) and LOW-FAT of <5 g/100 g (mean 1.1±1.29 g/100 g). Similarly, HIGH-CARB was defined as >15 g/100 g carb (48.57±21.13 g/100 g) and LOW-CARB as <15 g/100 g (4.59±4.59 g/100 g). We aimed for the contents to be as far as possible away from the cut-off to provide the best distinction.

Images will be identical in size, controlled for low-level visual features (like luminance) and will be presented on the computer monitor during EEG recording sessions. Immediately after the presentation of each image, women

have to decide if the presented picture is a food or non-food object by pressing a button 1 or 2 on a response box. Accuracy and reaction time of behavioural responses will be collected. At the beginning of the experimental session, the participant will be presented with the instructions and will have to perform some trials (seven pictures of food and three of non-food) to familiarise herself with the task. Continuous 128-channel EEG will be recorded on the head surface. VEPs will be calculated separately for each image category (HIGH-FAT/LOW-CARB; HIGH-FAT/HIGH-CARB; LOW-FAT/LOW-CARB; LOW-FAT/HIGH-CARB). The preparation for the EEG recording will take approximately 30–45 min, while EEG sessions will last maximum 55 min. When the EEG experiment is completed, the patient will move to the preparation room for the validation task. Twenty pictures of food (five pictures from each food category) will be presented and the patient will be asked to rate how much she wants this food. During visit 2, four blocks with randomised images of food and non-food objects will be presented in a different order than during visit 1 to minimise potential sources of bias.

### Heart rate and heart rate variability
During EEG and the Auracle testing, we will monitor the participant's heart rate and heart rate variability to assess the level of arousal of the autonomic system. We will perform baseline and recovery measurements (2 min each) before and after EEG and the Auracle testing. For this purpose, we will use Actiheart, a compact chest-worn monitoring tool that apart from heart rate, measures inter beat interval. Specific time points before and after the test will be identified by a stopwatch. To analyse the data, we will use the commercial Actiheart Software and Kubios.

### Body composition
Body composition will be measured by bioimpedance (BIA 101, Akern, Italy) at both visits. Using four electrodes (two on the right hand and two on the right foot, at a distance of 3–4 cm from each other) and low-voltage electric flows, we will measure the resistance (Rz) and reactance (Xc) of tissue to assess the fat and lean mass. We will also use BIA InBody s10 to measure body composition by bioimpedance at both visits. The body composition analysis of InBody S10 is derived from the four-compartment model, which include Total Body Water, Protein, Mineral and Body Fat.[54]

### Eating behaviour ('Auracle') and choice of snacks
The Auracle (see figure 2) will be used as an objective measure of eating behaviour during snack session (see figure 3). Auracle is an eating-recognition system that measures vibrations generated from movement, chewing and teeth grinding. When worn, the contact microphone is positioned on the mastoid and when a person eats, vibrations from movement, chewing and grinding of teeth sensed by the device are transmitted and stored on an

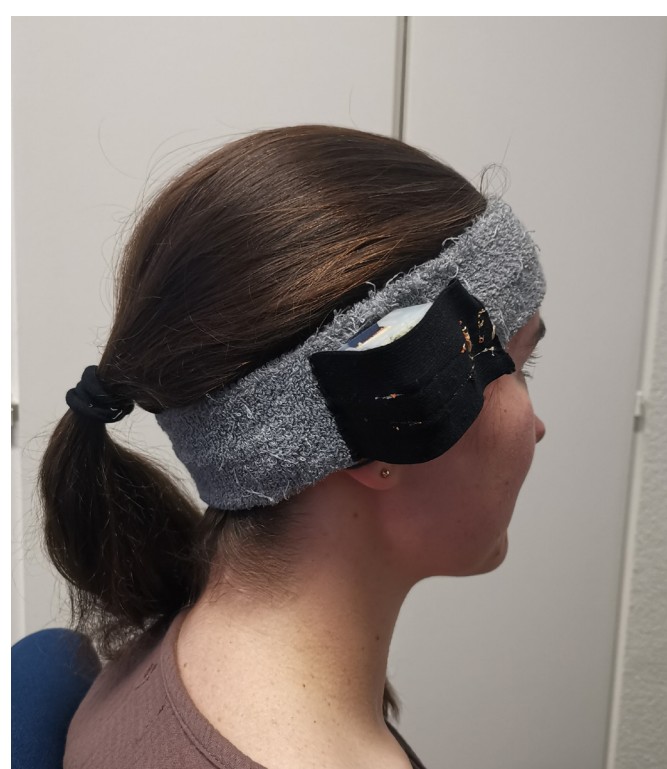

**Figure 2** The Auracle: eating recognition system.

embedded memory card. This device automatically identifies the eating rate and rapidity as well as the duration of food intake and other related dynamics with an accuracy exceeding 92%.[55] Auracle provides a holistic, detailed and accurate characterisation of objective correlates of eating behaviour. All of the sensors in Auracle (ADXL362 accelerometer and CM-01B microphone) are commercial, off-the-shelf devices. The device includes an MSP430 microcontroller (Texas Instruments, Dallas, Texas, USA), a push button and buzzer for user interaction. It is comfortable to wear because it is embedded in an elastic headband or a flexible plastic frame and possesses no risk to the mother or fetus. The Auracle device has been validated and has been used in healthy non-pregnant subjects.[55 56]

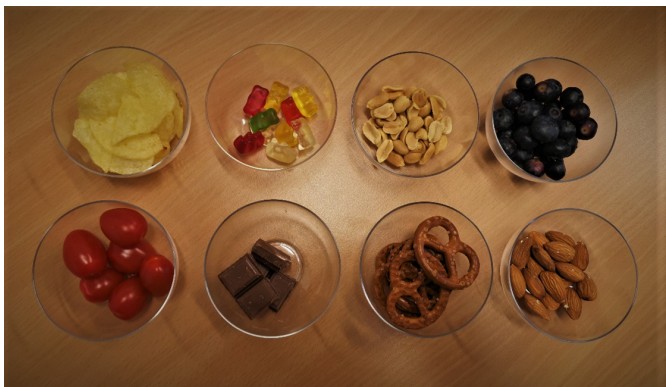

**Figure 3** Eight snacks of different carbohydrate and fat content.

Eight different crunchy snacks of different carbohydrate and fat content (two of each corresponding to one of the four categories described in the EEG part) will be offered (figure 3). As eating behaviour will be measured with the 'Auracle' device, crunchy-textured snacks provide a stronger, more easily detected vibrational signal. Before the snack session, the patient will be asked the second time about their current level of hunger (1—not at all to 5—very hungry) and fill out the 'PANAS' questionnaire. The weight of the different snacks will be measured before and after the patient has finished eating to determine the number of kcal and the quantity of fat, protein and carbohydrates eaten. Furthermore, the time interval between the first and last bite will be also measured using the 'Auracle' device and a stopwatch. We will also ask the woman questions regarding her subjective perceived eating rate (perceived eating duration, perceived eating speed and current eating speed compared with usual eating speed).

### Collection of blood samples
Capillary blood will be collected to measure HbA1c level. HbA1c will be analyzed by Afinion.

### Glucose sensors
Women with GDM will use Freestyle Libre 1 glucose sensors (Abbott) for flash glucose measurement for 14 days. Sensors will be applied on the back of the arm, with a sterile fibre being inserted under the skin into the interstitial fluid. Patients use a dedicated application LibreLink and will have to scan the glucose sensor at least every 8 hours to transfer data. Data will be linked in a coded form to the online platform Libreview (https://www.libreview.com/). During pregnancy, results will be discussed with their healthcare provider during the visits. For the post partum visit, patients will receive feedback by phone.

### Questionnaires related to eating behaviour and mental health
Women with and without GDM will fill out the same questionnaires during both visits (visits 1 and 2). Most women will fill out the questionnaires in French and a few in English, as around 80% of patients that participate in our studies are French-speaking. The following questionnaires will be filled out online before the study visits:

The French IES-2:[57] Validated self-report questionnaire that assesses individuals' tendency to follow their physical, hunger and satiety cues in determining when, what and how much to eat. The original version (English) consists of four subscales, Eating for Physical rather than Emotional Reasons (EPR, eight items), Reliance on Hunger and Satiety Cues (RHSC, six items), Unconditional Permission to Eat (UPE, six items) and the Body-Food Choice Congruence (three items) subscales. The validated French version, however, consists of 18 items divided into three subscales, the EPR (eight items), RHSC (six items) and the UPE (four items) subscales,[58] and has also demonstrated good psychometric properties.

In this study, we will assess intuitive eating with only the EPR and the RHSC subscales (14 items) of the French-adapted version of IES-2 and the English version. Scores of IES-2 range between 1 and 5. A higher score of the EPR subscale reflects eating as an answer to hunger and a lower score means eating to cope with emotional distress, whereas a higher score on the RHSC subscale signifies trust in internal cues, and a lower score reflects less ability to regulate food intake.

The Three-Factor Eating Questionnaire-Revised 18 (TFEQ-R18): Eighteen-item self-report questionnaire that measures three aspects of eating behaviours including cognitive restraint (six items), uncontrolled eating (nine items) and emotional eating (three items). Items are scored on a Likert scale ranging from 1 'almost always' to 4 'never'. A short, revised version of the Stunkard and Messick TFEQ was developed by Karlsson *et al*.[59] It has been also tested in the general population in France.[60]

Food Craving Questionnaire-Trait-reduced:[61] Fifteen-item self-report questionnaire for the assessment of the frequency and intensity of food craving. It consists of five subscales: lack of control over eating (five items), thoughts or preoccupation with food (five items), intentions and plans to consume food (two items), emotions before or during food craving (two items) and cues that may trigger food craving (one item). Items are scored on a Likert scale from 1 'never/not applicable' to 6 'always'. Higher scores reflect food-cue elicited craving. The French version was validated and has shown good psychometric proprieties.[62]

Edinburgh Postnatal Depression Scale (EPDS):[63] Ten-item self-report questionnaire to examine postnatal depression symptoms over the previous week, which are scored on a 4-point Likert scale. Scores range from 0 to 30. Higher scores indicate a higher risk of postnatal depression. The French version of the EPDS demonstrated good psychometric properties.[64]

Edinburgh Handedness Inventory short form:[65] This short (four items) and simple questionnaire measures the preference to use one hand more than the other. The French version has been previously used in research.[66]

During the study visits, participants will complete the following questionnaires:

The Positive and Negative Affective Schedule questionnaire (PANAS):[67] Self-report questionnaire that assesses mood or emotion during the last month. It consists of 20 items, 10 items measuring positive affect (eg, enthusiastic, proud) and 10 items measuring negative affect (eg, scared, ashamed), which are scored on a Likert scale from 1 'not at all' to 5 'extremely'. Individual scores on the PANAS range from 10 to 50, with higher scores representing higher levels of positive affect. The validated French version has shown good psychometrics.[68]

The Perceived Stress Scale (PSS-10): Ten-item questionnaire widely used to assess stress levels during the last month. Responses are rated on a Likert scale from 0 'never' to 5 'very often'. The results range from 0 to 40, with higher scores indicating higher perceived stress. The

French version of PSS-10 has shown good psychometric properties.[69]

The Food Craving Questionnaire-State:[70] Fifteen-item questionnaire measures the intensity of momentary food cravings. Items are scored on a 5-point scale ranging from 'strongly disagree' to 'strongly agree'. Higher scores indicate more intense current food cravings. The French version was translated and used in previous research by the research team of our collaborator Ulrike Toepel.[66]

The current hunger level scale: This is one item developed for this study assessing the current perceived hunger level using a Likert scale ranging from 1 'just a little/not at all' to 5 'very much'.

The food wanting questionnaire: Twenty-item questionnaire was developed for this study and will be completed after the EEG (see Behavioural task/EEG). Women will be shown 20 pictures of food and asked 'How much do you want this food?'. They will then have to indicate their response on a Likert scale from 0 'not at all' to 10 'very much'.

## Data management and statistical analyses
### Statistical analysis plan

The sample size is based on previous EEG studies of similar design,[23 25] but all analyses are exploratory, as to our knowledge so far no such study has been conducted during pregnancy or the post partum period. Statistical analyses will be performed with STATA V.15.1 (2017) and MATLAB V.9.5 (R2018b). Processing of the EEG data will be performed using Cartool software and statistical software.[52] Demographic and other descriptive variables will be represented as means and SD or percentages where appropriate. Analysis of variance, t-tests or $\chi^2$ tests will be used to determine group differences between responses to food and non-food pictures, different food categories, brain responses between pregnancy and post partum states, between women with and without GDM and between normal weight and overweight/obese women. Regression analyses will be used to model the relationships between predictor variables and outcomes (such as between brain responses to food viewing with intuitive eating and metabolic health such as weight, body fat and glucose control). In the second step, we will investigate the impact of covariates, where appropriate, such as maternal age, gestational age and prepregnancy BMI on outcomes and adjust for them if necessary.

Twenty women with valid data for the primary endpoint in each group (n=40) will be investigated during pregnancy and at 6 months postpartum. In order to achieve this, we had to include more patients.

## ETHICS AND DISSEMINATION

The local ethical committee approved the study protocol (study number 2021-01976). The study poses little to no risk to participants. Signed informed consent will be obtained from all participating women. The results of the study will be disseminated through national and international conferences, in peer-reviewed journals and presented at public conferences and other scientific meetings.

## Significance and outlook

The prevalence of obesity continues to increase worldwide. This study seeks to understand brain reactions to the viewing of food images and their changes during pregnancy and in the post partum period. Food cravings and disordered eating behaviours include implicit processes in the brain. Studying these processes could help unravel mechanistic pathways associated with weight gain. Our study may provide novel evidence of scientific and clinical importance to understand eating behaviour, potentially opening perspectives for future interventions that alter reactions to energy-dense food in general or at specific food nutrients and/or brain regions to manage excessive GWG, weight retention, and to reduce adverse metabolic outcomes in the post partum period. It is a potential opportunity to reduce craving, incentive sensitisation and thus the overstimulation of reward circuits by food cues in pregnant and post partum states where a dynamic change in craving and weight gain is observed. The study aims to provide data for a better understanding and long-lasting interventions that can influence the health of the mother and the child.

### Author affiliations
[1]Obstetric Service, Department Woman-Mother-Child, Lausanne University Hospital and University of Lausanne, Lausanne, Switzerland
[2]Faculty of Biology and Medicine, University of Lausanne, Lausanne, Switzerland
[3]Service of Endocrinology, Diabetes and Metabolism, Lausanne University Hospital and University of Lausanne, Lausanne, Switzerland
[4]Laboratory for Investigative Neurophysiology (the LINE), Radiology Department, Lausanne University Hospital and University of Lausanne, Lausanne, Switzerland
[5]The Sense Innovation and Research Center, Lausanne and Sion, Switzerland
[6]CIBM Center for Biomedical Imaging, Lausanne, Switzerland
[7]Thayer School of Engineering, Dartmouth College, Hanover, New Hampshire, USA
[8]Department of Surgery, Geisel School of Medicine, Dartmouth College, Hanover, New Hampshire, USA
[9]Institute of Higher Education and Research in Healthcare (IUFRS), University of Lausanne, Lausanne, Switzerland
[10]Neonatology service, Department Woman-Mother-Child, Lausanne University Hospital and University of Lausanne, Lausanne, Switzerland

**Contributors** JJP designed the study with input from all coauthors, particularly DYQ. CR, UT and MMM contributed to the creation and design of the EEG section, along with SS, DYQ, AL and JJP. SS was responsible for the nutritional part regarding the lunch and snacks. AH contributed to the psychology part and AL-S, along with DYQ, to the statistical and data analysis part. RJH was responsible for the design of 'Auracle', adapting it together with DYQ. AL-S wrote the first draft and all authors critically revised the final version of the manuscript.

**Funding** This work was supported by a donation from the Dreyfus foundation and an unrestricted educational grant from NovoNordisk. JJP's research is funded by the Leenaards price in translational medical research (2019, Grant Number 4522.8). RJH's research programme is supported by the US National Science Foundation under award numbers CNS-1565269, CNS-1835983, CNS-1565268 and CNS-1835974.

**Competing interests** None declared.

**Patient and public involvement** Patients and/or the public were involved in the design, or conduct, or reporting, or dissemination plans of this research. Refer to the Methods section for further details.

**Patient consent for publication** Obtained.

**Provenance and peer review** Not commissioned; externally peer reviewed.

**ORCID iDs**
Anna Lesniara-Stachon http://orcid.org/0000-0001-7063-1308
Alain Lacroix http://orcid.org/0000-0001-8835-5110
Antje Horsch http://orcid.org/0000-0002-9950-9661
Jardena J Puder http://orcid.org/0000-0002-0460-7614

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
