## [Reviewer comments · BMJ Open]

ARTICLE DETAILS

TITLE (PROVISIONAL)	Brain responses to food viewing in women during pregnancy and postpartum and their relationship with metabolic health: Study protocol for the FOODY Brain Study, a prospective observational study.
AUTHORS	Lesniara-Stachon, Anna; Quansah, Dan Yedu; Schenk, Sybille; Retsa, Chrysa; Halter, Ryan J; Murray, Micah; Lacroix, Alain; Horsch, Antje; Toepel, Ulrike; Puder, J

VERSION 1 – REVIEW

REVIEWER	Smith, Taryn University of California Davis, Institute for Global Nutrition
REVIEW RETURNED	20-Oct-2022

GENERAL COMMENTS	This is a well written and interesting protocol describing a study that utilises a novel approach of measuring brain responses to food cues to understand food choices and eating behaviours during pregnancy and postpartum and the relationship to women's metabolic health. The justification and significance of the study is well explained. I have just a few minor comments: 1. What is the rationale for only inviting a subset (n = 14) of the women at 6 months postpartum? I understand that drop-out rates may be high in the postpartum period when women have a young infant, but to maximise sample size wouldn't all women be invited?2. Be consistent with spelling of behaviour – it is spelt behaviour and behavior inconsistently throughout the manuscript. Similarly behavioural and behavioral.3. Introduction line 50, define GWG abbreviation at first use.4. Is there a trial status update that could be added? When are the anticipated start and end dates? How long is it anticipated that it will take to enrol the 20 healthy and 20 women with GDM?5. Has the trial been registered at a registry such as clinicaltrials.gov or similar?
--

REVIEWER	Thompson, Katherine A. The University of North Carolina at Chapel Hill
REVIEW RETURNED	29-Nov-2022

GENERAL COMMENTS	The current manuscript is a protocol which describes a study examining brain responses to food viewing in perinatal women. In addition to brain responses, the study will also evaluate eating
--

behaviors, food cravings, and metabolic health markers. Overall, the topic of the study is interesting and important for understanding eating behaviors and metabolic health during a vulnerable period – the perinatal period. However, there are there are also some concerns regarding the organization and flow of the introduction and the justification of the aims. Specific comments and suggestions are outlined by section below:

Introduction:

- Overall, the introduction reads as choppy in a way that makes the justification of the aims difficult to follow. Consider moving the section on metabolic challenges of pregnancy to the end of the introduction before the aims. A suggested flow of the introduction would be focusing on: (1) metabolic health (what markers are important and why), (2) how eating behaviors relate to metabolic health (food cravings, overeating, etc.), (3) how the brain relates to both eating behaviors and metabolic health, and (4) why this is important to study in the perinatal period. This would help set up the aims.

- The paragraphs do not have enough interpretation of prior literature. Currently, the sentences are summaries of research results that make it difficult for the reader to understand why this topic is important (when it is). For example: the paragraph that starts with “Response to food viewing in regions linked to cognitive evaluation processes....” Does not have an explanation for why these studies are important to discuss. Why does it matter that brain activation is different in individuals with overweight versus controls? How does this relate to the aims of the study? Consider doing a thorough review of the whole introduction to expand on these questions.

- The first sentence on the second page that starts “The few existing clinical data show that reward...” is long and difficult to follow. Consider breaking this up into shorter sentences.

- The phrase “emotional eaters’ is slightly misleading and stigmatizing. Since emotional eating is not a disorder, a person cannot be diagnosed with clinical emotional eating. Consider re-wording this to people who engage in more frequent emotional eating or something else that reflects the methods of the study cited.

- The introduction does not clearly state why perinatal women are an important population to study. Consider expanding upon this point. How do eating behaviors change during the perinatal period? What impact do pregnancy-specific metabolic factors have on long-term health? This is an incredibly important topic, however, the introduction does not clearly state why.

- Aim 3 states that the study will investigate differences in brain responses to food pictures in relationship to different emotional states and stress level. This aim is confusing. Neither emotional states nor stress are mentioned in the introduction. There is no discussion of why stress or emotional states may be important in relation to brain responses to food pictures – particularly during the perinatal period. Please clarify the aim and include these topics in the introduction.

- The introduction is missing hypotheses. Please include for each aim.

Methods:

- In general, the EEG methods are beyond my expertise. Consider having the manuscript reviewed by someone familiar with EEG.

	- Eating rate is included as a secondary outcome. This is not a typical eating behavior variable in the disordered eating field. Please explain why it is included in this study. There is no mention of eating rate in the introduction, so it is unclear how it relates to the overall aims of the study. - Why were only a subgroup of 14 women invited to participate in the postpartum visit? Please explain. How will these women be selected? Is it random or using some inclusion/exclusion criteria? - In the description of the Auracle device to assess eating rate, please include validation information. Has it been used in previous studies for this purpose? - The questionnaires will be completed in French and “a few in English.” Which questionnaires will be in English? Why was this decided? How will the study assess English language proficiency among participants? Please expand on this.
--	---

VERSION 1 – AUTHOR RESPONSE

Reviewer: 1

Dr. Taryn Smith, University of California Davis

Comments to the Author:

This is a well written and interesting protocol describing a study that utilises a novel approach of measuring brain responses to food cues to understand food choices and eating behaviours during pregnancy and postpartum and the relationship to women’s metabolic health. The justification and significance of the study is well explained. I have just a few minor comments:

Dear Dr. Taryn Smith,

Thank you very much for reviewing our manuscript. We have carefully considered your comments in revising the manuscript. Below are our responses directly below your comments.

1. What is the rationale for only inviting a subset (n = 14) of the women at 6 months postpartum? I understand that drop-out rates may be high in the postpartum period when women have a young infant, but to maximise sample size wouldn’t all women be invited?

Ad 1. Thank you very much for this comment. Indeed, in response to the reviewers’ comments and also initial methodological problems to have valid EEG data, we changed our study protocol and we decided to include all participating women for both visits. In addition, we needed to increase the recruited sample size both of women with GDM and healthy women. This was due to the issues of non-valid EEG data when we started analyzing them with our EEG expert collaborators (eg too much noise, low quality of signal, small general movements of the patients). We aim to have a total of 40 women (20 women with GDM and 20 healthy controls), all with valid EEG data during the pregnancy visit that also plan to attend the postpartum visit. We know understood the reasons for the non-valid EEG data and took all the precautions to completely reduce or at least minimize the chance to have any more non-valid EEG measures.

2. Be consistent with spelling of behaviour – it is spelt behaviour and behavior inconsistently throughout the manuscript. Similarly behavioural and behavioral.

Ad 2. Thank you for this remark. This was corrected throughout the manuscript to behaviour/behavioural.

3. Introduction line 50, define GWG abbreviation at first use.

Ad 3. Thank you, we defined this abbreviation.

4. Is there a trial status update that could be added? When are the anticipated start and end dates? How long is it anticipated that it will take to enrol the 20 healthy and 20 women with GDM?

Ad 4. Thank you. In the methods section, we added information on dates: "The study started in March 2022 and is scheduled to end in June 2024." In December 2022, we started a second visits for the patients who were recruited in March 2022. But as we need 20 women with valid EEG in each group during pregnancy, we need to prolong the recruitment phase (analyzing all existing EEG data this will amount to 33 women with GDM and 32 healthy controls). We aim complete the recruitment phase at the end of April.

5. Has the trial been registered at a registry such as clinicaltrials.gov or similar?

Ad 5. Thank you for this comment. This study has not been registered in the registry as it is not a trial and only an observational study. That is why we decided to publish a Methods paper, also to clarify our objectives, population and measures.

In addition, we expanded and changed structure of introduction, based on the comments of the other reviewer. We have extended the inclusion criteria to 36 weeks gestation (instead of 32 weeks) due to the large number of women in good health with advanced pregnancy wishing to participate in the study.

Reviewer: 2

Katherine A. Thompson, The University of North Carolina at Chapel Hill

Comments to the Author:

The current manuscript is a protocol which describes a study examining brain responses to food viewing in perinatal women. In addition to brain responses, the study will also evaluate eating behaviors, food cravings, and metabolic health markers. Overall, the topic of the study is interesting and important for understanding eating behaviors and metabolic health during a vulnerable period – the perinatal period. However, there are there are also some concerns regarding the organization and flow of the introduction and the justification of the aims. Specific comments and suggestions are outlined by section below:

Dear Dr Katherine A. Thompson,

Thank you very much for reviewing our manuscript. We have carefully considered all your comments in revising the manuscript. Please find a detailed response to each of your comments below.

Introduction:

- Overall, the introduction reads as choppy in a way that makes the justification of the aims difficult to follow. Consider moving the section on metabolic challenges of pregnancy to the end of the introduction before the aims. A suggested flow of the introduction would be focusing on: (1) metabolic health (what markers are important and why), (2) how eating behaviors relate to metabolic health (food cravings, overeating, etc.), (3) how the brain relates to both eating behaviors and metabolic health, and (4) why this is important to study in the perinatal period. This would help set up the aims. Thank you very much for this valuable comment. We expanded and completely changed the structure of introduction based on your suggestion and agree that this has improved it a lot. The new subsection include: "Metabolic health and eating behaviours", "Brain regions in relationship to food cues", "Stress, emotions and eating behavior", "Pregnancy", "Postpartum period" and "Electroencephalography".

- The paragraphs do not have enough interpretation of prior literature. Currently, the sentences are summaries of research results that make it difficult for the reader to understand why this topic is important (when it is). For example: the paragraph that starts with "Response to food viewing in regions linked to cognitive evaluation processes...." Does not have an explanation for why these studies are important to discuss. Why does it matter that brain activation is different in individuals with

overweight versus controls? How does this relate to the aims of the study? Consider doing a thorough review of the whole introduction to expand on these questions.

Thank you for this comment. We have added an interpretation of the previous literature and decided to slightly revised the objective of our study. In the second aim, we specified metabolic health outcomes at which we especially we will look at such as weight, body fat and glucose. Therefore, it will be of interest to us to compare the brain's response to the sight of food also between overweight/obese and normal-weight women.

- The first sentence on the second page that starts "The few existing clinical data show that reward..." is long and difficult to follow. Consider breaking this up into shorter sentences.

Thank you. We have divided this sentence into several shorter sentences.

- The phrase "emotional eaters" is slightly misleading and stigmatizing. Since emotional eating is not a disorder, a person cannot be diagnosed with clinical emotional eating. Consider re-wording this to people who engage in more frequent emotional eating or something else that reflects the methods of the study cited.

Thank you for this comment, we agree and we changed it according to your suggestion.

- The introduction does not clearly state why perinatal women are an important population to study. Consider expanding upon this point. How do eating behaviors change during the perinatal period? What impact do pregnancy-specific metabolic factors have on long-term health? This is an incredibly important topic, however, the introduction does not clearly state why.

Thank you very much for this comment. In the introduction, we added information about the perinatal period, the change in eating behaviour and its relation to the metabolic health of both mother and infant. We have expanded on this in two subsections: pregnancy and the postnatal period.

- Aim 3 states that the study will investigate differences in brain responses to food pictures in relationship to different emotional states and stress level. This aim is confusing. Neither emotional states nor stress are mentioned in the introduction. There is no discussion of why stress or emotional states may be important in relation to brain responses to food pictures – particularly during the perinatal period. Please clarify the aim and include these topics in the introduction.

Thank you for this comment. In the introduction, we have completed the information on stress and emotions and their relationship to eating behaviour, metabolic health and brain responses in subsection : "Stress, emotions and eating behavior" as well as in the 2 chapters related to "Pregnancy" and "Postpartum".

- The introduction is missing hypotheses. Please include for each aim.

Thank you for this suggestion. We included the hypothesis at the end of the introduction, before the aims.

Methods:

- In general, the EEG methods are beyond my expertise. Consider having the manuscript reviewed by someone familiar with EEG.

Thank you for this important suggestion. At the level of study design, we collaborated with EEG experts such as our co-authors of this manuscript (Micah M. Murray, Ulrike Toepel and Chrysa Retsa). They also have worked in the field of EEG and nutrition, weight and eating behaviour (especially Ulrike Toepel). We had also contacted other experts such as Camille Cr  z  .

- Eating rate is included as a secondary outcome. This is not a typical eating behavior variable in the disordered eating field. Please explain why it is included in this study. There is no mention of eating rate in the introduction, so it is unclear how it relates to the overall aims of the study.

It is correct that it is not a typical eating behavior variable. However, eating rate can be regarded as an aspect of eating behavior. Increased eating rate (eg. the amount of food intake in grams per unit of time (minute) and measured in calories) has been associated with obesity. We added a few sentences in the introduction about the relationship between an increased eating rate, energy intake and a higher risk of obesity.

- Why were only a subgroup of 14 women invited to participate in the postpartum visit? Please explain. How will these women be selected? Is it random or using some inclusion/exclusion criteria? Thank you for this comment. In response to the reviewers' comments and also initial methodological problems to have valid EEG data, we changed our study protocol and we decided to invite all the women to both visits. Also, we decided to increase the sample size both of women with GDM and healthy women, due to the issues of non-valid EEG data when we started analyzing them with our EEG expert collaborators (eg too much noise, low quality of signal, small general movements of the patients). We aim to have a total of 40 women (20 women with GDM and 20 healthy controls), all with valid EEG data during the pregnancy visit that also plan to attend the postpartum visit. We know understood the reasons for the non-valid EEG data and took all the precautions to completely reduce or at least minimize the chance to have any more non-valid EEG measures.

- In the description of the Auracle device to assess eating rate, please include validation information. Has it been used in previous studies for this purpose?

Thank you for this comment. Yes, Auracle device was already used for this purpose and validated in previous studies (1,2). We added this now in the MS.

Auracle gives an exhaustive assessment of eating behavior and can provide an empirical evidence of real-time eating behavior measures. It can sense, process, and classify audio data in real time, detects eating activities with an accuracy exceeding 92%. It also distinguishes eating from other activities, detects and stores eating behavior measures (1,2). Auracle provides a holistic, detailed and accurate characterization of objective correlates of eating behavior.

So far, for your information, we observed in some preliminary analysis that 60% of objective eating behavior measures (duration and rate) were explained by subjective self-report measures regarding duration during pregnancy ($r=0.78$). We also observed that global metrics across entire eating period detected by Auracle correlated with lower weight ($r=-0.61$), and lower gestational weight gain ($r=-0.72$) in women during pregnancy.

1. Bi S, Wang T, Davenport E, Peterson R, Halter R, Sorber J, Kotz D. Toward a wearable sensor for eating detection. In: WearSys 2017 - Proceedings of the 2017 Workshop on Wearable Systems and Applications, co-located with MobiSys 2017. 2017.

2. Bi S. Detection of health-related behaviors using head-mounted devices. Dartmouth College; 2021)

- The questionnaires will be completed in French and "a few in English." Which questionnaires will be in English? Why was this decided? How will the study assess English language proficiency among participants? Please expand on this.

Thank you for this comment. We have all the questionnaires in both language versions (French and English). Based on the experience from previous studies, around 80% of our patients are French-speaking, so they complete all the questionnaires in French. Some of our patients are not French-speaking, if they are English-speaking they can also participate in our study and then they fill in all the questionnaires in English. Being able to understand either French or English orally and in writing is our inclusion criteria for the study and this is assessed during recruitment process.

We agree that this may have been misleading, so we have added a short explanation: "Most women will fill out the questionnaires in French and a few in English, as around 80% of our patients are French-speaking."

Also in the inclusion criteria we added a sentence: "Women in both groups should be able to fluently speak and write either French or English."

In addition to that, we have extended the inclusion criteria to 36 weeks gestation (instead of 32 weeks) due to the large number of women in good health with advanced pregnancy wishing to participate in the study.

VERSION 2 – REVIEW

REVIEWER	Smith, Taryn University of California Davis, Institute for Global Nutrition
REVIEW RETURNED	09-Feb-2023
GENERAL COMMENTS	Thank you for the responses and revisions. I have no further comments.